# Metformin Affects Serum Lactate Levels in Predicting Mortality of Patients with Sepsis and Bacteremia

**DOI:** 10.3390/jcm8030318

**Published:** 2019-03-06

**Authors:** Fu-Cheng Chen, Chia-Te Kung, Hsien-Hung Cheng, Chi-Yung Cheng, Tsung-Cheng Tsai, Sheng-Yuan Hsiao, Chien-Hung Wu, Chih-Min Su

**Affiliations:** 1Departments of Emergency Medicine, Kaohsiung Chang Gung Memorial Hospital, Chang Gung University College of Medicine, Kaohsiung 83301, Taiwan; fuchang@cgmh.org.tw (F.-C.C.); kungchiate@gmail.com (C.-T.K.); kendrew@cgmh.org.tw (H.-H.C.); qzsecawsxd@cgmh.org.tw (C.-Y.C.); takk921@yahoo.com.tw (T.-C.T.); kmu9100027@cgmh.org.tw (S.-Y.H.); mic@cgmh.org.tw (C.-H.W.); 2School of Medicine, Chung Shan Medical University, Kaohsiung 80424, Taiwan

**Keywords:** sepsis, metformin, lactate, mortality

## Abstract

This study determined if the use of metformin affected the prognostic value of hyperlactatemia in predicting 28-day mortality among patients with sepsis and bacteremia. We enrolled adult diabetic patients with sepsis and bacteremia. Of 590 patients, 162 and 162 metformin users and nonusers, respectively, were selected in propensity matching. The mean serum lactate levels in metformin users were higher than those in nonusers (4.7 vs. 3.9 mmol/L, *p* = 0.044). We divided the patients into four groups based on quick Sepsis-related Organ Failure Assessment (qSOFA) scores. No significant difference was found among nonusers with qSOFA score <2, nonusers with qSOFA score ≥2, and metformin users with qSOFA score <2. The lactate levels in metformin users with qSOFA score ≥2 were higher than those in other groups, and significant differences were found in both nonsurvivors (8.9 vs. 4.6 mmol/L, *p* = 0.027) and survivors (6.4 vs. 3.8 mmol/L, *p* = 0.049) compared with metformin users with qSOFA score <2. The best cut-off point to predict 28-day mortality in metformin users (5.9 mmol/L; area under the receiver operating characteristic curve (AUROC), 0.66; 95% confidence interval (CI), 0.55–0.77) was higher than that in nonusers (3.6 mmol/L; AUROC 0.63; 95% CI, 0.56–0.70). Metformin users had higher lactate levels than nonusers in increasing sepsis severity. Serum lactate levels could be useful in predicting mortality in patients using metformin, but higher levels are required to obtain more precise results.

## 1. Introduction

Sepsis is a time-dependent disease, especially if it rapidly progresses to severe sepsis or septic shock [1]. The key element in the treatment of septic patients is early recognition and appropriate interventions. Serum lactate is a good biomarker for the risk stratification and monitoring of sepsis treatment [2,3]. Elevated serum lactate levels are strongly associated with mortality in patients with severe sepsis [4]. Clinically, serum lactate is a useful biomarker to risk-stratify patients with severe sepsis presenting to the emergency department (ED) [5].

However, several diseases or mechanisms can also cause hyperlactatemia, complicating the interpretation of serum lactate levels [6]. Among these complicating factors, metformin use is often encountered. Metformin remains the optimal drug for monotherapy in patients with diabetes because of its inexpensiveness, proven safety, and potential benefits on cardiovascular outcomes, compared with insulin and sulfonylureas [7]. Metformin is a biguanide drug that interferes with mitochondrial metabolism and inhibits hepatic uptake of serum lactate [8].

Because the use of metformin may affect serum lactate levels, it could also interfere with the prognostic value of lactate levels in sepsis. Some previous studies have indicated that metformin could elevate the serum lactate level and affect its prognostic value [9,10], but some studies have shown that metformin use was not associated with hyperlactatemia [11,12]. The aim of the present study was to determine whether metformin use affects the prevalence of hyperlactatemia and the prognostic value of serum lactate level in predicting the 28-day mortality in patients with sepsis and bacteremia.

## 2. Material and Methods

### 2.1. Study Design

This was a single-center retrospective observational study on diabetic patients admitted to the ED with sepsis, at least one set of blood culture tests, and serum lactate level test. The data were retrieved from electronic medical records (EMRs) between 1 January 2007 and 31 December 2013 from Kaohsiung Chang Cheng Memorial Hospital—a 2692-bed acute-care teaching hospital, the largest medical center in Southern Taiwan providing both primary and tertiary referral care. The Institutional Review Board of Chang Cheng Memorial Hospital approved this study (IRB No.: 103-0053B.). Patient informed consent was waived.

### 2.2. Study Setting and Population

All the patients from whom blood culture samples were collected in the ED were screened in a computer database, as shown in Figure 1. Patients with diabetes mellitus (DM) and bacteremia who had been administered a serum lactate level test were enrolled in the study. A ratio of 1:1 matched control group patients were selected by propensity score matching. For all enrolled patients, the following data were collected, retrospectively: demographic characteristics, preexisting major comorbidities, initial vital signs and laboratory tests results, major infection source, and microorganisms isolated from the blood cultures.

### 2.3. Measurement and Data Collection

Preexisting major comorbidities, including type 2 DM and infection site, were obtained from the International Statistical Classification of Diseases and related health problems, 9th Revision coding [13].

Serum lactate levels were initially measured within the six hours that sepsis was suspected. Serum lactate (mmol/L) levels were measured using a serum-based immunoassay (Unicel DxC 880i Synchron, Beckman Coulter Inc., Brea, CA, USA).

Medication history was reviewed, and metformin use was defined as the continuous administration of the drug at a dosage of >1000 mg/day for at least two weeks before sepsis and bacteremia developed.

### 2.4. Definitions

Sepsis was diagnosed when provider-suspected infection was reported in the EMR and two or more systemic inflammatory response syndrome (SIRS) criteria (using initial vital signs and laboratory examinations obtained at the ED) were met [14]. The SIRS score included a temperature of >38 °C or <36 °C, heart rate of >90 beats/min, respiratory rate of >20 cycles/min, and abnormal white blood cell count (>12,000/µL or <4000/µL or >10% immature (bands) forms).

Blood culture samples with potential contaminating pathogens (e.g., coagulase-negative *Staphylococcus*, *Propionibacterium acnes*, *Micrococcus*, *Corynebacterium* spp., and *Peptostreptococcus*) were considered contaminations and were regarded as “no bacteremia” [15,16].

Septic shock is defined as sepsis-induced hypotension persisting despite adequate fluid resuscitation [17]. Hypotension is defined by a systolic arterial pressure <90 mmHg, mean arterial pressure <60 mmHg, or a reduction in systolic blood pressure of <40 mmHg from baseline, despite adequate volume resuscitation, in the absence of other causes of hypotension [18].

We used the quick Sepsis-related Organ Failure Assessment (qSOFA) score to evaluate the severity of illness (using first ED vital signs). The qSOFA score included systolic blood pressure of ≤100 mmHg, respiratory rate of ≥22 cycles/min, and Glasgow Coma Scale score of ≤13. Each of the above conditions was equivalent to one point, and the score ranged from 0 to 3 [19].

The major outcome was defined as 28-day mortality.

### 2.5. Statistical Analysis

Statistical analyses were performed using IBM SPSS Statistics for Windows, version 24 (IBM Corp., Armonk, NY, USA). Continuous variables were expressed as mean ± standard deviation and were analyzed using the *t*-test. Categorical variables were expressed as numbers and percentages and were compared using the χ^2^ test or Fisher exact test. We used the propensity score adjustment method to adjust for patient imbalance between metformin users and nonusers using variables including age, sex, comorbidities, source of infection, septic shock, and qSOFA score. Balance was achieved satisfactorily based on mean standardized differences and variance ratios of the two groups. Age, sex, comorbidities, source of infection, septic shock, qSOFA score, and serum lactate levels were included in multivariate backward logistic regression analyses to identify variables independently associated with 28-day mortality. The performance of the serum lactate levels was assessed by the area under the receiver operating characteristic curve (AUROC) to describe the diagnostic accuracy of the scores. The best cut-off point was calculated by using the Youden index [20]. *p*-values < 0.05 were considered statistically significant.

## 3. Results

### 3.1. Baseline Characteristics

Among the 590 patients with diabetes with sepsis and bacteremia, 162 and 162 metformin users and nonusers, respectively, were selected in propensity matching. The baseline characteristics of all patients and matched patients by propensity score are shown in Table 1. The propensity score-matching method resulted in balanced groups with underlying baseline characteristics. Table 1 shows that before propensity matching, patients with liver cirrhosis and chronic renal insufficiency were less likely to take metformin because these conditions were contraindications to the use of this medicine [21]. No significant difference was found in the 28-day mortality between the two groups (18% vs. 21%; *p* = 0.328). Patients taking metformin had a higher incidence of urinary tract infection (50% vs. 41%, *p* = 0.043) and higher serum lactate level (4.7 mmol/L vs. 3.8 mmol/L; *p* = 0.009) than metformin nonusers. After propensity matching, metformin users had only a higher serum lactate level (4.7 mmol/L vs. 3.9 mmol/L; *p* = 0.044) than nonusers.

### 3.2. Multivariate Logistic Regression Analyses for 28-Day Mortality

Multivariate logistic regression models were used to assess the influence of variables on the 28-day mortality in Table 2. Before propensity matching, serum lactate was a significant factor in affecting the 28-day mortality (adjusted odds ratio (AOR), 1.09; 95% confidence interval (CI), 1.03–1.15, *p* = 0.003), and other significant factors included age, chronic renal insufficiency, malignancy, respiratory tract infection, urinary tract infection, septic shock, and qSOFA score ≥2. After propensity matching, serum lactate was still a significant factor affecting the 28-day mortality (AOR, 1.09; 95% CI, 1.01–1.18, *p* = 0.023), and other significant factors included malignancy, respiratory tract infection, urinary tract infection, and septic shock.

### 3.3. Subgroup Analysis by qSOFA Score

We used qSOFA in performing the subgroup analysis and found that the mean serum lactate level of metformin users with qSOFA score ≥2 (7.3 mmol/L) was much higher than that of the other three groups (4.2 mmol/L in nonusers with qSOFA score ≥2, *p* < 0.001; 3.9 mmol/L in metformin users with qSOFA score <2, *p* = 0.001; 3.8 mmol/L in nonusers with qSOFA score <2, *p* < 0.001) (Figure 2a). We also found that the mean serum lactate level in metformin users with qSOFA score ≥2 was higher than that in metformin users with qSOFA score <2 in both nonsurvivors (8.9 vs. 4.6 mmol/L, *p* = 0.027) and survivors (6.4 vs. 3.8 mmol/L, *p* = 0.049) (Figure 2b). However, no significant difference was found between nonusers with qSOFA score ≥2 and nonusers with qSOFA score <2 in both nonsurvivors (5.2 vs.4.3 mmol/L, *p* = 0.584) and survivors (3.9 vs. 3.7 mmol/L, *p* = 0.756).

### 3.4. Receiver Operating Characteristic Curve Analysis for Lactate in Predicting 28-Day Mortality

The receiver operating characteristic curve analysis was performed to evaluate the discriminative ability of serum lactate level for 28-day mortality (Figure 3). The AUROC of serum lactate level in all patients (0.63; 95% CI, 0.57–0.69), metformin nonusers (0.63; 95% CI, 0.56–0.70), and metformin users (0.66; 95% CI, 0.55–0.77) in predicting the 28-day mortality is shown in Table 3. Different serum lactate level cut-offs resulted in different sensitivities and specificities. In addition to the conventional serum lactate level, we also used Youden index to calculate the best cut-off point in all patients (lactate = 3.6 mmol/L, sensitivity = 0.58, specificity = 0.64), metformin nonusers (lactate = 3.6 mmol/L, sensitivity = 0.54, specificity = 0.69), and metformin users (lactate = 5.9 mmol/L, sensitivity = 0.49, specificity = 0.85).

### 3.5. Microbiology Results of Blood Culture

Analysis of culture microbiology results revealed that Gram-negative bacteria accounted for most of the isolates (213/324, 65.7%), whereas Gram-positive bacteria represented only one-third of all pathogens (111/324, 34.3%). No statistically significant survival difference was found among these different patterns of blood culture isolates. The three most common pathogens were *Escherichia coli* (*n* = 100, 30.9%), *Klebsiella pneumoniae* (*n* = 54, 16.7%), and *Staphylococcus aureus* (*n* = 77, 10.1%).

## 4. Discussion

In the present study, the serum lactate level in patients with sepsis and bacteremia was significantly higher in metformin users; however, this difference did not result in differences in the mortality rate between the groups. In addition, the difference in serum lactate level in metformin users increased with increasing sepsis severity. The results of our study demonstrated that metformin did not affect the use of serum lactate levels in predicting mortality if a higher cut-off value was selected.

Several studies have investigated the association between metformin use and serum lactate level in patients with infection or sepsis. Green et al. conducted a retrospective cohort study of 1947 patients with SIRS and suspected infection (192 and 1755 (only 343 DM cases) metformin users and nonusers with a 28-day mortality rate of 8% and 17%, respectively) and demonstrated that hyperlactatemia was associated with an increased adjusted 28-day mortality risk among nonmetformin users but not among metformin users [9]. Metformin users had a lower mortality rate (8% vs. 17%) than nonusers in their study, which may have affected the prediction ability of serum lactate level. In our study, we found that the higher cut-off value of serum lactate level was needed for predicting mortality in metformin users.

Doenyas-Barak et al. conducted a retrospective cohort study of 162 patients with serum lactate level >10 mmol/L and septic shock (44 and 118 metformin users and nonusers (only 23 DM cases) with in-hospital mortality rates of 56.8% and 88.1%, respectively) and demonstrated that high lactate concentration with low mortality rate was found in those taking metformin [10]. We found that the serum lactate level in metformin users was much higher than that in nonusers if patients had higher disease severity. Owing to their selection criteria of serum lactate level >10 mmol/L, metformin users may have more mild cases than nonusers, and metformin users consequently had a lower mortality rate.

Park et al. conducted a retrospective cohort study with propensity matching of 213 patients with severe sepsis or septic shock (71 and 142 metformin users and nonusers with 28-day mortality rate of 10% and 9%, respectively) that demonstrated that the serum lactate levels in metformin users were initially elevated in the early phase of resuscitation, and then no significant difference was found over the initial 24-h period [11]. Although the outcome of patients with severe sepsis has improved in the past few decades, the mortality rate remains higher (15% to 30%) [22,23]. Their study included patients with severe sepsis or septic shock, but the mortality rate was lower than usual (only 9–10%), possibly because the disease severity of patients was low and the serum lactate level was not significantly different between metformin users and nonusers.

Lee et al. conducted a cross-sectional study with 1954 patients with type 2 diabetes (745 metformin users and 1200 nonusers) and demonstrated that the use of metformin was not associated with hyperlactatemia in patients with type 2 diabetes [12]. Their study found that metformin did not affect hyperlactatemia in patients with diabetes (not specific to sepsis or infection) generally. Our study also found no significant difference in the serum lactate level between metformin users with qSOFA score <2 and nonusers. Metformin use may not have a significant effect in patients with mild infection, but once the disease severity increases (qSOFA score >2), the difference in serum lactate level will be noticeable.

The causality between metformin use and the development of hyperlactatemia remains controversial. The proposed mechanism includes inhibition of gluconeogenesis and mitochondrial impairment [24]. When combined with sepsis, the risk of developing hyperlactatemia may increase significantly because both conditions affect lactate metabolism. The results of the present study indicated that metformin resulted in a higher increase in serum lactate levels in metformin users with qSOFA score ≥2 compared with those with qSOFA score <2 (both survivors and nonsurvivors), but no such difference was found between nonusers. The qSOFA is a new measure that provides simple criteria to identify adult patients with infection who are likely to have poor outcomes [5,19]. The elevation of serum lactate level in metformin users with qSOFA score ≥2 may be due to the rapid disease progression leading to multiple organ failure or septic shock, resulting in increased lactate production and decreased metabolism. In addition, metformin use may lead to other synergistic effects causing hyperlactatemia. However, detailed pathophysiologic process and causality could not be explained by our study, and further investigation will be needed.

Our study has certain limitations. First, because this was a retrospective study, serum lactate levels were not measured for all septic patients in the ED. Those whose serum lactate levels were measured tended to have increased sepsis severity stage. This meant that the patients included in our analysis included those with a higher disease severity. Second, we had only one data for lactate level, and peak levels were not measured. However, in our sepsis protocol, those whose serum lactate levels were measured would receive aggressive treatment, and most of them had a decrease in lactate level. Third, the Sepsis-3 definition uses qSOFA for suspected sepsis, but we found the sensitivity of qSOFA was lower than SIRS, and qSOFA was more suitable to classify severity in our previous study [25]. Besides, qSOFA was not the criteria for sepsis in our research years. Therefore, we used SIRS criteria to define sepsis and used the qSOFA score to evaluate the severity of illness. Finally, we categorized patients as metformin users and nonusers but did not measure actual serum metformin levels to definitively prove the correlation of metformin and serum lactate level.

## 5. Conclusions

In patients with sepsis and bacteremia, metformin users had a higher prevalence of hyperlactatemia than nonusers as sepsis severity increased. Serum lactate levels still could be used to predict 28-day mortality in such patients, but the precision was best at higher levels.

## Figures and Tables

**Figure 1 jcm-08-00318-f001:**
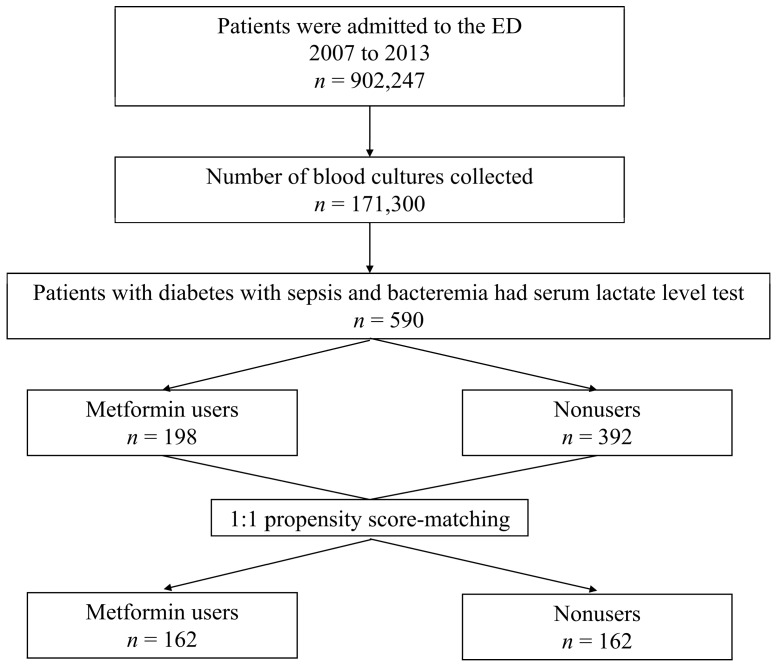
Flow diagram of the study design and analyses. ED: emergency department.

**Figure 2 jcm-08-00318-f002:**
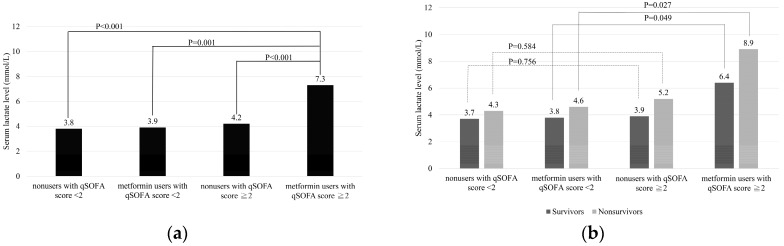
Subgroup analysis by qSOFA score. (**a**) The mean serum lactate level in metformin users with qSOFA score ≥2 was significantly higher than that in other groups. Significant difference at *p* < 0.001 compared with nonusers with qSOFA score ≥2 and nonusers with qSOFA score <2; Significant difference at *p* = 0.001 compared with metformin users with qSOFA score <2. (**b**) The mean serum lactate level was significantly higher in metformin users with qSOFA score ≥2 than in metformin users with qSOFA score <2 in both nonsurvivors (*p* = 0.027) and survivors (*p* = 0.049). No significant difference was found in nonusers with qSOFA score ≥2 compared with nonusers with qSOFA score <2 in both nonsurvivors (*p* = 0.584) and survivors (*p* = 0.756).

**Figure 3 jcm-08-00318-f003:**
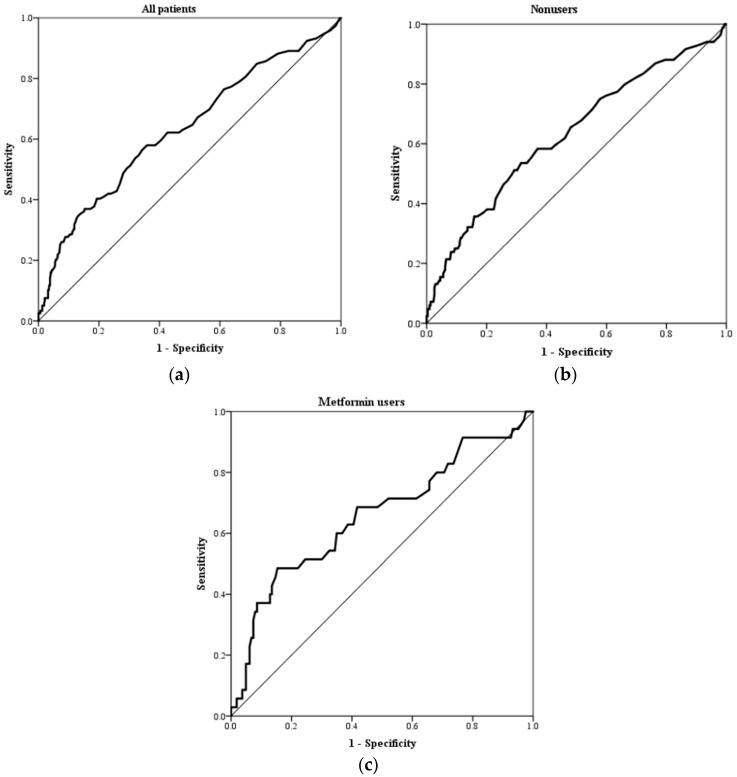
Receiver operating characteristic curve analysis of serum lactate levels for predicting 28-day mortality: (**a**) all patients, (**b**) nonusers, and (**c**) metformin users.

**Table 1 jcm-08-00318-t001:** Demographics and clinical characteristics of all patients and propensity–matched patients.

Variable	All Patients (*n* = 590)			Propensity-Matched Patients (*n* = 324)		
	Metformin Users (*n* = 198)	Nonusers (*n* = 392)	*p*-Value	Metformin Users (*n* = 162)	Nonusers (*n* = 162)	*p*-Value
Age, years	70 ± 12	69 ± 13	0.363	69 ± 12	69 ± 13	0.935
Male sex, *n* (%)	87 (44%)	188 (48%)	0.451	65 (40%)	75 (46%)	0.313
Comorbidities						
Liver cirrhosis	7 (4%)	44 (11%)	0.002 ^a^	4 (3%)	5 (3%)	1.000
Chronic renal insufficiency	42 (21%)	120 (31%)	0.019 ^a^	37 (23%)	40 (25%)	0.794
Congestive heart failure	12 (6%)	35 (9%)	0.262	12 (7%)	10 (6%)	0.826
Malignancy	26 (13%)	71 (18%)	0.128	24 (15%)	24 (15%)	1.000
Suspected infection focus						
Respiratory tract	64 (32%)	131 (33%)	0.853	50 (31%)	48 (30%)	0.904
Urinary tract	98 (50%)	159 (41%)	0.043 ^a^	75 (46%)	77 (48%)	0.911
Skin and soft tissue	14 (7%)	36 (9%)	0.436	11 (7%)	12 (7%)	1.000
Intra–abdomen	23 (12%)	57 (15%)	0.374	21 (13%)	24 (15%)	0.748
Others	35 (18%)	88 (22%)	0.198	34 (21%)	27 (17%)	0.394
Septic shock	46 (23%)	95 (24%)	0.838	31 (19%)	33 (20%)	0.889
qSOFA score ≥2	54 (27%)	105 (27%)	0.922	40 (25%)	37 (23%)	0.794
28-day mortality	35 (18%)	84 (21%)	0.328	29 (18%)	27 (17%)	0.883
Serum lactate (mmol/L)	4.7 ± 4.1	3.8 ± 3.4	0.009 ^a^	4.7 ± 4.3	3.9 ± 2.9	0.044 ^a^
C-reactive protein (mg/L)	154.2 ± 125.4	148.2 ± 113.3	0.579	155.1 ± 125.8	143.3 ± 112.	0.397
WBC (1000/mm^3^)	13.6 ± 7.4	14.9 ± 12.8	0.214	13.6 ± 7.5	14.2 ± 8.0	0.491
BUN (mg/dL)	40.5 ± 33.1	41.5 ± 34.3	0.772	41.9 ± 35.1	38.4 ± 29.2	0.410
Creatinine (mg/dL)	1.8 ± 1.4	2.4 ± 2.3	0.002 ^a^	1.8 ± 1.5	2.1 ± 1.9	0.291

qSOFA: quick Sepsis-related Organ Failure Assessment; WBC: white blood cell; BUN: blood urea nitrogen. ^a^
*p* < 0.05.

**Table 2 jcm-08-00318-t002:** Multiple logistic regression analyses for variables associated with 28-day mortality.

Variable	All Patients (*n* = 590)				Propensity-Matched Patients (*n* = 324)			
	COR (95% CI)	*p*-Value	AOR (95% CI)	*p*-Value	COR (95% CI)	*p*-Value	AOR (95% CI)	*p*-Value
Age (+1 year)	1.02 (1.01–1.04)	0.037 ^a^	1.02 (0.99–1.04)	0.063	1.00 (0.97–1.03)	0.990	-	-
Male sex	1.14 (0.70–1.86)	0.595	-	-	0.93 (0.44–1.98)	0.859	-	-
Liver cirrhosis	1.13 (0.52–2.46)	0.754	-	-	3.64 (0.63–21.02)	0.149	-	-
Chronic renal insufficiency	1.92 (1.16–3.19)	0.011 ^a^	1.98 (1.21–3.23)	0.007 ^a^	1.84 (0.84–4.06)	0.129	-	-
Congestive heart failure	0.67 (0.29–1.56)	0.353	-	-	0.44 (0.10–1.90)	0.274	-	-
Malignancy	2.59 (1.49–4.54)	0.001 ^a^	2.67 (1.54–4.62)	<0.001 ^a^	4.51 (1.96–10.41)	<0.001 ^a^	4.64 (2.13–10.08)	<0.001 ^a^
Respiratory tract infection	2.31 (1.16–4.59)	0.018 ^a^	2.18 (1.37–3.46)	0.001 ^a^	4.56 (1.54–13.50)	0.006 ^a^	3.55 (1.82–6.90)	<0.001 ^a^
Urinary tract infection	0.42 (0.22–0.79)	0.008 ^a^	0.39 (0.24–0.65)	<0.001 ^a^	0.50 (0.18–1.35)	0.169	0.44 (0.21–0.93)	0.032 ^a^
Skin and soft tissue infection	1.28 (0.51–3.25)	0.603	-	-	0.79 (0.15–4.15)	0.781	-	-
Intra-abdomen infection	0.76 (0.31–1.87)	0.546	-	-	0.80 (0.19–3.41)	0.759	-	-
Other infection	1.12 (0.46–2.73)	0.800	-	-	1.74 (0.45–6.70)	0.422	-	-
Septic shock	3.22 (1.96–5.29)	<0.001 ^a^	3.21 (1.97–5.24)	<0.001 ^a^	3.48 (1.58–7.64)	0.002 ^a^	3.70 (1.80–7.61)	<0.001 ^a^
qSOFA score ≥2	1.62 (0.98–2.67)	0.058	1.69 (1.04–2.78)	0.035 ^a^	1.56 (0.70–3.44)	0.275	-	-
Metformin use	0.79 (0.47–1.32)	0.371	-	-	1.01 (0.50–2.02)	0.987	-	-
Serum lactate (+1 mmol/L)	1.09 (1.03–1.16)	0.003 ^a^	1.09 (1.03–1.15)	0.003 ^a^	1.07 (0.99–1.16)	0.093	1.09 (1.01–1.18)	0.023 ^a^

AOR: adjusted odds ratio; CI: confidence interval; COR: crude odds ratio; qSOFA: quick Sepsis-related Organ Failure Assessment. ^a^
*p* < 0.05.

**Table 3 jcm-08-00318-t003:** Receiver operating characteristic analysis of serum lactate levels to predict 28-day mortality.

Outcome	Group	AUC	95% CI	*p*	Cut–Off Point	Sensitivity	Specificity	Youden Index
28-day mortality	All patients	0.63	0.57–0.69	<0.001 ^a^	2.0	0.81	0.31	0.12
					3.6 ^b^	0.58	0.64	0.22
					4.0	0.51	0.69	0.21
	Nonusers	0.63	0.56–0.70	<0.001 ^a^	2.0	0.79	0.34	0.14
					3.6 ^b^	0.54	0.69	0.22
					4.0	0.48	0.73	0.21
	Metformin users	0.66	0.55–0.77	0.003 ^a^	2.0	0.83	0.26	0.09
					4.0	0.60	0.63	0.23
					5.9 ^b^	0.49	0.85	0.33

ROC: receiver operating characteristic; AUC: area under the curve; CI: confidence interval. ^a^
*p* < 0.05. ^b^ Optimal cut-off point.

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
