# Peer review of "Metformin Affects Serum Lactate Levels in Predicting Mortality of Patients with Sepsis and Bacteremia"

_jcm, 2019, doi:10.3390/jcm8030318_

Round 1
Reviewer 1 Report
The manuscript, “metformin affects serum lactate levels in predicting mortality of patients with sepsis and bacteremia” compares serum lactate levels in diabetic patients with sepsis due to bacteremia between metformin users and non-users. In addition, the authors evaluate whether high lactate levels in metformin users can predict 28d mortality. There are a number of concerns, however:
1. Minor point, but the authors start the introduction stating that “sepsis is a time-dependent disease in the emergency department”. However, sepsis can present on any unit of the hospital, so this phrase should be changed.
2. The authors only screen patients for positive blood cultures instead of evaluating all positive cultures. This may significantly limit the number of patients with sepsis and bias the data.
3. Since this is retrospective data, it is not clear that the authors were able to confirm that patients were actually taking the metformin prior to presentation. There is a statement that “medication history was reviewed, and metformin use was defined as the continuous administration…for at least two weeks before sepsis”. How were the authors able to confirm the patients were taking the metformin?
4. It is not clear whether the lactate levels analyzed were the initial 6h lactate or the peak lactate levels. Could the authors clarify this? If the peak lactate levels were not assessed, this will significantly bias the data as well.
5. Another very minor point: There is a “%” missing from Table 1 in the first “nonusers” column.
6. The authors defined sepsis using SIRS criteria, which is not in concordance with the current Sepsis-3 definition. This is a limitation of the manuscript and should be discussed.
Author Response
Response to Reviewer 1 Comments
Point 1: Minor point, but the authors start the introduction stating that “sepsis is a time-dependent disease in the emergency department”. However, sepsis can present on any unit of the hospital, so this phrase should be changed. 

Response 1: Thank you for your valuable and constructive advice. Following your recommendations, we have removed "in the emergency department" and revised this portion as follows: "Sepsis is a time-dependent disease, especially if it rapidly progresses to severe sepsis or septic shock." (page: 1, lines: 32-33).
Point 2: The authors only screen patients for positive blood cultures instead of evaluating all positive cultures. This may significantly limit the number of patients with sepsis and bias the data.
Response 2: Thank you for your comment. We chose positive blood culture instead of all positive cultures because the false positives of other body fluid cultures are more likely. Perhaps more patients will be included in the study, but selection bias will occur. Although our study includes fewer patients, we believe that patients with sepsis with bacteremia are representative of the entire sepsis population, and it is appropriate to use this group of patients to discuss the use of metformin.
Point 3: Since this is retrospective data, it is not clear that the authors were able to confirm that patients were actually taking the metformin prior to presentation. There is a statement that “medication history was reviewed, and metformin use was defined as the continuous administration…for at least two weeks before sepsis”. How were the authors able to confirm the patients were taking the metformin?
Response 3: Thank you for your comment. We use electronic medical records to check drug use. If the patient takes metformin, there must be a record in the electronic file. According to the onset time, we confirmed that the patient had a record of using metformin at least two weeks before the onset of sepsis and bacteremia, which was regarded as continuous administration.
Point 4: It is not clear whether the lactate levels analyzed were the initial 6h lactate or the peak lactate levels. Could the authors clarify this? If the peak lactate levels were not assessed, this will significantly bias the data as well.
Response 4: Thank you for your valuable and insightful comments. We had only one data for lactate level, and peak level were not measured. However, in our sepsis protocol, those whose serum lactate levels were measured would receive aggressive treatment, and most of them had a decrease in lactate level. This is a limitation of the study and we have added in the discussion. (page: 9, lines: 247-250).
Point 5: Another very minor point: There is a “%” missing from Table 1 in the first “nonusers” column.
Response 5: Thank you for your correction. We have added a “%”in the first “nonusers” column in Table 1.
Point 6: The authors defined sepsis using SIRS criteria, which is not in concordance with the current Sepsis-3 definition. This is a limitation of the manuscript and should be discussed.
Response 6: Thank you for your valuable and constructive advice. The Sepsis-3 definition use qSOFA for suspect sepsis, but we found the sensitivity of qSOFA was lower than SIRS and qSOFA is more suitable to classify severity in our previous study [1]. Beside, qSOFA were not the criteria for sepsis in our research years. Therefore, we used SIRS criteria to definite sepsis and used qSOFA score to evaluate the severity of illness. This is a limitation of the study and we have added in the discussion. (page: 9, lines: 250-253)
1. Chen FC, Kung CT, Cheng HH, Cheng CY, Tsai TC, Hsiao SY, et al. Quick Sepsis-related Organ Failure Assessment predicts 72-h mortality in patients with suspected infection. Eur J Emerg Med 2018.
Reviewer 2 Report
Review comment sheet 1 A brief summary The paper describes that metformin users had higher lactate levels than nonusers in increasing sepsis severity. Serum lactate levels could be useful in predicting mortality in patients using metformin, but higher levels are required to obtain more precise results. This is a nice paper. However, I have some comments. 2 Overall evaluation The findings from this paper are excellent and novel regardless of a review. 3 Main problem This manuscript contained some questions described below. I think this paper has very interesting, this study contributes to future's clinical medicine largely. I have some questions from a point of view of clinical medicine. In this paper the authors focused metformin and serum lactate in sepsis, I would like to know the relationship serum lactate and lactic acidosis. What do the authors think about that ? Maybe fetal factor in sepsis is lactic acidosis due to circulation and/or respiratory failure finally. And I want to know that what is the reason of high values of lactate in septic metformin users. Do the authors describe the baseline data of serum lactate in status before sepsis in metformin users ? I think that it is important how far the lactic acid value has risen from baseline due to sepsis I want to know the other biomarkers in blood or urine, for example serum CRP WBC BUN Cre and so on, urinary L-FABP and so on.
Author Response
Response to Reviewer 2 Comments
Point 1: In this paper the authors focused metformin and serum lactate in sepsis, I would like to know the relationship serum lactate and lactic acidosis. What do the authors think about that ? Maybe fetal factor in sepsis is lactic acidosis due to circulation and/or respiratory failure finally.
Response 1: First of all, thank you for your appreciation of our research. Regarding the relationship between serum lactate and lactic acidosis, our thoughts are like your opinion. The production of lactic acidosis could be caused by circulation and/or respiratory failure, and this may also be one of the causes of the increase in serum lactate level of metformin users once the disease severity increases.
Point 2: And I want to know that what is the reason of high values of lactate in septic metformin users. Do the authors describe the baseline data of serum lactate in status before sepsis in metformin users ? I think that it is important how far the lactic acid value has risen from baseline due to sepsis I want to know the other biomarkers in blood or urine, for example serum CRP WBC BUN Cre and so on, urinary L-FABP and so on.
Response 2: Thank you for your valuable and constructive advice. We didn’t have baseline data of serum lactate in status before sepsis in metformin users. So we can't provide data from baseline to sepsis. We compare the data of CRP WBC BUN Cre in Table 1. The only difference between the two groups is that the creatinine of metformin users is better than nonusers before propensity matching. We think that is because metformin has traditionally been regarded as contraindicated in chronic kidney disease. Besides, we didn’t have urinary L-FABP data.
Round 2
Reviewer 1 Report
All reviewer comments were sufficiently addressed
Reviewer 2 Report
Review comment sheet
In this paper, answers are properly and carefully described in my question, and it has been corrected well.